# The NLRP1 Inflammasome in Human Skin and Beyond

**DOI:** 10.3390/ijms21134788

**Published:** 2020-07-06

**Authors:** Gabriele Fenini, Tugay Karakaya, Paulina Hennig, Michela Di Filippo, Hans-Dietmar Beer

**Affiliations:** 1Department of Dermatology, University Hospital of Zurich, 8091 Zurich, Switzerland; Gabriele.Fenini@usz.ch (G.F.); Tugay.Karakaya@usz.ch (T.K.); Paulina.Hennig@usz.ch (P.H.); Michela.DiFilippo@usz.ch (M.D.F.); 2Faculty of Medicine, University of Zurich, 8091 Zurich, Switzerland

**Keywords:** NLRP1, CARD8, inflammasome, skin, keratinocyte, inflammation

## Abstract

Inflammasomes represent a group of protein complexes that contribute to host defense against pathogens and repair processes upon the induction of inflammation. However, aberrant and chronic inflammasome activation underlies the pathology of numerous common inflammatory diseases. Inflammasome assembly causes activation of the protease caspase-1 which in turn activates proinflammatory cytokines and induces a lytic type of cell death termed pyroptosis. Although NLRP1 (NACHT, leucine-rich repeat and pyrin domain containing 1) was the first inflammasome sensor, described almost 20 years ago, the molecular mechanisms underlying its activation and the resulting downstream events are incompletely understood. This is partially a consequence of the poor conservation of the NLRP1 pathway between human and mice. Moreover, recent evidence demonstrates a complex and multi-stage mechanism of NLRP1 inflammasome activation. In contrast to other inflammasome sensors, NLRP1 possesses protease activity required for proteolytic self-cleavage and activation mediated by the function-to-find domain (FIIND). CARD8 is a second FIIND protein and is expressed in humans but not in mice. In immune cells and AML (acute myeloid leukemia) cells, the anti-cancer drug talabostat induces CARD8 activation and causes caspase-1-dependent pyroptosis. In contrast, in human keratinocytes talabostat induces NLRP1 activation and massive proinflammatory cytokine activation. NLRP1 is regarded as the principal inflammasome sensor in human keratinocytes and UVB radiation induces its activation, which is believed to underlie the induction of sunburn. Moreover, gain-of-function mutations of *NLRP1* cause inflammatory skin syndromes and a predisposition for the development of skin cancer. SNPs (single nucleotide polymorphisms) of *NLRP1* are associated with several (auto)inflammatory diseases with a major skin phenotype, such as psoriasis or vitiligo. Here, we summarize knowledge about NLRP1 with emphasis on its role in human keratinocytes and skin. Due to its accessibility, pharmacological targeting of NLRP1 activation in epidermal keratinocytes represents a promising strategy for the treatment of the numerous patients suffering from NLRP1-dependent inflammatory skin conditions and cancer.

## 1. Introduction

Skin represents the essential outer barrier of the human body and is in permanent contact with the environment [1]. Its outermost compartment, the epidermis, consists of several layers of densely packed cells, named keratinocytes, in defined stages of differentiation. The epidermis is a constantly renewing tissue. Under homeostatic conditions, proliferation of keratinocytes is restricted to the innermost layer. These basal keratinocytes migrate outwards, differentiate and change their expression profile, finally die and fuse to a structure, termed cornified envelope, which is particularly important for the barrier function of the epidermis. In contrast to the underlying dermis, a connective tissue, the epidermis contains only few immune cells [2,3]. Keratinocytes of the epidermis do not only form the outer structural barrier, but express in a constitutive or inducible manner also molecules, peptides and proteins, which are actively involved in anti-bacterial and anti-viral defense as well as in immune responses, such as anti-microbial peptides or proinflammatory cytokines [3].

In the skin, but also in other tissues, the detection of stress factors can mount a highly complex response, termed inflammation [4]. Inflammation is an essential and beneficial process required to combat pathogenic invaders or to induce tissue repair after injury. However, when chronic or/and out of control, inflammation becomes detrimental and causes or contributes to numerous widespread (inflammatory) diseases. Therefore, the interest in understanding the molecular events regulating inflammation is very high and these efforts can potentially result in the development of novel strategies for the pharmacological targeting of inflammation. Chronic inflammation constitutes also a hallmark in cancer development [5]. By creating a “smoldering” inflammatory environment, innate and adaptive immune cells can trigger the development and growth of tumors as well as invasion and metastasis [6]. In contrast, via anti-tumor cytotoxic T cell activity and cytokine-mediated lysis of tumor cells, adaptive immune cells inhibit the growth of tumors, and activation of the immune system by immunotherapy represents a successful anti-cancer strategy. Therefore, inflammation represents a double-edged sword for carcinogenesis, able to support this process at all stages and at the same time be capable of efficient suppression [6].

## 2. Inflammasomes

Upon detection of different stress factors, termed pathogen- or damage/danger-associated molecular patterns (PAMPs, DAMPs), cytoplasmic NOD-like receptors (NLRs) form oligomers. These complexes, termed inflammasomes, cause a series of downstream events and ultimately inducing inflammation [7,8]. In 2002, the group of Jürg Tschopp described the activation of NLRP1, the first inflammasome sensor [9]. Later, the NLRC4 [10], NLRP3 [11,12] as well as the AIM2 (absent in melanoma 2) [13,14] and pyrin [15] inflammasomes were discovered. Direct binding of cytoplasmic DNA, which may derive from pathogens or dying cells, activates the AIM2 inflammasome and contributes to defense against bacteria and viruses [13,14]. In contrast, the molecular mechanisms underlying NLRP3 activation are not well understood. NLRP3 is activated by many different stress factors, which cannot all bind to NLRP3 and cause its activation, due to their chemical and structural diversity. Therefore, indirect mechanisms of NLRP3 activation are discussed [16]. As it is believed that NLRP3 activation contributes to common inflammatory diseases but is dispensable for immunity against pathogens, therapeutic targeting and inhibition of NLRP3 is a major issue for pharmacological treatment of the many patients suffering from NLRP3-associated diseases [17,18]. Most inflammasome sensors contain a pyrin domain [19,20,21] which is, like CARD (caspase activation and recruitment domain), characterized by a death domain fold, allowing homotypic protein-protein interactions [22]. After stress factor-induced activation and oligomerization of NLRs, they bind to the pyrin domain of the adaptor protein ASC (apoptosis-associated speck-like protein containing a CARD) via homotypic interaction inducing the formation of large multimers, termed ASC specks [23]. Via its CARD, ASC interacts in a homotypic manner with (pro-)caspase-1, resulting in dimerization of the protease followed by proteolytic self-activation. Caspase-1 represents the effector molecule of inflammasomes and belongs to a family of cysteine proteases, which regulate cell death and apoptosis [22]. The proinflammatory cytokines proIL(interleukin)-1β and -18 are substrates of caspase-1 [24]. Active caspase-1 cleaves and thereby activates proIL-1β and -18 and the secretion of these active cytokines induces inflammation. Interestingly, IL-1β and -18 as well as several other proteins with extracellular functions, particularly during inflammation, lack a signal sequence for their secretion by the canonical endoplasmic reticulum/Golgi-dependent pathway. These proteins leave the cell via regulated or passive mechanisms collectively termed unconventional protein secretion [25,26]. Interestingly, caspase-1 activity is not only required for activation of proIL-1β and -18 but also for their secretion and release [27]. Caspase-1 cleaves gasdermin D and the resulting aminoterminal fragment inserts into the extracellular membrane upon oligomerization [28]. The resulting pores allow release of IL-1β, IL-18 and other cytoplasmic proteins, as well as the entrance of water, causing swelling and finally bursting of the cells [28,29,30,31,32]. However, living cells are also able to secrete IL-1β with low levels of gasdermin D activation [33,34,35,36,37,38,39]. To summarize, inflammasomes represent a group of protein complexes, which induce inflammation upon sensing different types of stress factors.

## 3. The Molecular Mechanisms Underlying NLRP1 Activation

Human NLRP1 is with 1473 amino acids the largest inflammasome sensor [40]. It possesses an aminoterminal pyrin domain, a NACHT domain, LRRs (leucine-rich repeats), FIIND (function-to-find domain), and a carboxyterminal CARD (Figure 1a). In contrast to NLRP3, AIM2, pyrin and some poorly characterized inflammasome sensors [40], the carboxyterminal CARD of NLRP1 is part of the effector domain, which induces ASC speck formation [41,42]. The function of the carboxyterminal pyrin domain, which is missing in murine NLRP1, is not clear but seems to be required for keeping NLRP1 in a self-inhibited state together with LRRs [40,43].

In contrast to the above-mentioned inflammasome sensors, NLRP1 has a FIIND. This domain possesses protease activity needed for self-cleavage of NLRP1 between phenylalanine 1212 and serine 1213 in the SF/S motif of the FIIND [48]. F1212 and S1213 are strictly required for NLRP1 activation, as well as the histidine residue H1186. Most likely, after deprotonation of S1213 supported by H1186 (and possibly by an unknown acidic amino acid acting as proton acceptor for H1186), S1213 represents nucleophile attacking the C of the C-N bond between F1212 and S1213. Binding the hydrophobic side chain of F1212 in a pocket of the FIIND would then position the backbone in a conformation suitable for a nucleophilic attack by S1213 [48]. Therefore, the self-cleaving protease activity of FIIND might be based on a classical catalytic triad commonly found in hydrolases, such as proteases [49]. NLRP1 self-cleavage is necessary but not sufficient for NLRP1 activation [41,48,50,51]. The FIIND consist of the two domains ZU5 (for ZO-1 and UNC5) and UPA (UNC5, PIDD and Ankyrins), which are present in other proteins such as PIDD1 (p53-induced protein with a death domain protein 1) or CARD8 (Figure 1b) [50]. Interestingly, PIDD1 undergoes self-processing via a HF/S motif between ZU5 and UPA and is able to induce NF-κB activation or caspase-2-dependent apoptosis [52]. After processing, ZU5 and UPA further interact and hold both domains bound to each other by non-covalent interactions [50,53,54]. 

Further insight into the mechanisms underlying NLRP1 activation comes from murine studies. In contrast to humans, mice express several NLRP1 alleles and particularly NLRP1b confers susceptibility to infection by *Bacillus anthracis* [40,55]. Lethal factor (LF) is a metalloproteinase and part of *B. anthracis’* lethal toxin. LF is able to cleave and inactivate mitogen-activated protein kinases of the host cells, thereby inhibiting the host defense system. Additionally, LF does also cleave murine NLRP1b in the aminoterminus causing its activation [56,57]. Although human NLRP1 is not cleaved and activated by LF, an engineered proteolytic cleavage event in the aminoterminus induces also human NLRP1 activation [58]. Surprisingly, activation of human and murine NLRP1 is strictly dependent on proteasomal activity demonstrating that proteolytic degradation is an essential step in NLRP1 activation [53,59,60,61,62]. This because after auto-proteolytic activation of NLRP1b, the carboxyterminal effector domain is inhibited by interaction with the aminoterminal NLRP1b fragment. After LF-induced cleavage of the aminoterminal part, the newly generated aminoterminus of NLRP1b is targeted by the N-end rule pathway, ubiquitinated and degraded by the proteasome [53,54,63,64]. After this “functional degradation” of the inhibitory aminoterminal fragment, the liberated carboxyterminal effector domain of NLRP1b induces ASC speck formation and activation of caspase-1 [53,54]. The intracellular bacterium *Shigella flexneri* activates several types of inflammasomes [65], including NLRP1b [66,67]. Interestingly, *S. flexneri* expresses an E3 ubiquitin ligase that ubiquitinates NLRP1b and causes its activation independently of the N-end rule pathway but dependent on the proteasome [54]. In conclusion, self-activation in the FIIND is a prerequisite for NLRP1 activation. Then, the carboxyterminal effector domain remains in an inhibited state due to its interaction with the aminoterminal NLRP1 fragment, until the latter is ubiquitinated and degraded by the proteasome (Figure 2a). One can expect that human NLRP1 is activated in a similar way as murine NLRP1b. Still, all the players and, most importantly, the involved E3 ubiquitin ligases remain to be identified. In contrast to all known inflammasomes sensors, NLRP1 is activated upon self-processing in the FIIND and partial proteasomal degradation, termed functional degradation.

## 4. Talabostat and CARD8

In addition to *B. anthracis* and *S. flexneri*, murine NLRP1b is also activated by the pathogens *Listeria monocytogenes* [67] and *Toxoplasma gondii* [68] as well as by the anti-cancer drug talabostat (Val-boroPro, PT-100) [51,69]. Talabostat inhibits members of the dipeptidyl peptidase (DPP) family, including DPP8 and DPP9, which are supposed to be involved in NLRP1b activation [69], but also quiescent cell proline dipeptidase (QPP) and FAP (fibroblast activating protein) [70]. The molecular mechanisms underlying caspase-1 activation by talabostat are poorly characterized but it is believed that a substrate of DPP8/DPP9 is involved in NLRP1b activation [69,71]. For talabostat-induced NLRP1b activation, FIIND processing is a prerequisite and the proteasome pathway is required [51,72]. Interestingly, talabostat induces mainly pyroptosis in an NLRP1b-dependent but ASC-independent manner and secretion of very low levels of mature IL-1β (if at all) [51,69,71,73]. In general, ASC-independent inflammasome activation is believed to induce mainly pyroptosis without major processing of proIL-1β [74,75]. Similarly, talabostat is only a very weak inducer of mature IL-1β secretion in human PBMCs and human THP-1 cells [69,71]. Surprisingly, induction of pyroptosis in THP-1 cells and other AML (acute myeloid leukemia) cells by talabostat is mediated by CARD8 rather than by NLRP1, demonstrating that CARD8 is also able to form a type of inflammasome (Figure 2b) [64,76].

CARD8 shares with NLRP1 not only the FIIND but also the carboxyterminal CARD and the self-cleavage at the SF/S motif (Figure 1b) [50,76]. CARD8 was originally identified as TUCAN (tumor-up-regulated CARD-containing antagonist of caspase nine), as it interacts with caspase-9 and inhibits Apaf1-dependent apoptosis [77], and as CARDINAL (CARD
inhibitor of NF-κB-activating ligands) that inhibits NF-κB activation [78]. In contrast, it was also reported that CARD8 induces apoptosis and interacts with caspase-1 regulating its activation [79]. Splicing of CARD8 leads to several variants (Figure 1b), which might have specific properties and functions. CARD8 was first described as a 49 kDa protein (T48 isoform) lacking exons 1-3 and only later the full-length 60 kDa isoform was described [80]. The 55 kDa isoform (T54) was shown to inhibit caspase-8 and -9 and it is specifically associated with cancer cells [81]. Interestingly, CARD8 also takes part in the regulation of the NLRP3 inflammasome [12]. It directly interacts with NLRP3 and dampens activation of the inflammasome sensor, which is relevant for different inflammatory diseases [82,83,84]. CAPS (cryopyrin-associated periodic syndrome) is an autoinflammatory disease characterized by recurrent systemic inflammation and caused by mutations in the NLRP3 gene [85]. Mutated NLRP3 in patients suffering from CAPS cannot interact with CARD8 which in turn cannot inhibit NLRP3, thus causing increased production of mature IL-1β [84]. PFAPA (periodic fever with aphthous stomatitis, pharyngitis, and cervical adenitis) is a quite common autoinflammatory condition, affecting children. These children express loss-of-function variants of CARD8 lacking the FIIND and CARD domains, hence unable to bind and inhibit NLRP3 [83]. Furthermore, a CARD8 missense mutant defective in binding of NLRP3 was identified in members of a family suffering from Crohn’s Disease (CD) [82]. Peripheral monocytes of these patients secrete increased levels of IL-1β upon NLRP3 activation and the patients respond to IL-1β inhibitors. Interestingly, the identified mutation is dominant-negative [82]. CARD8 is also linked to CD due to its interaction and negative regulation of NOD2 (nucleotide-binding oligomerization domain-containing protein 2) [86].

Although talabostat induces CARD8-dependent pyroptosis in human AML cells, such as THP-1 cells, and NLRP1b-dependent pyroptosis in murine immune cells [51,76], human keratinocytes treated with the drug secrete very high levels of mature IL-1β [71]. Although human primary keratinocytes (HPKs) express also high levels of CARD8 (Fenini, unpublished), talabostat-induced IL-1β secretion requires expression of NLRP1 and ASC [71] (Fenini, unpublished). It is not yet understood why NLRP1 senses talabostat in human keratinocytes, whereas the drug activates CARD8 in THP-1 cells, as both cell types express NLRP1 and CARD8.

## 5. The NLRP1 Inflammasome in Human Keratinocytes

In 2001, NLRP1 was identified as a novel Ced-4 family member and termed DEFCAP (Death Effector Filament-forming Ced-4-like Apoptosis Protein) [87] or NAC (nucleotide-binding domain and CARD) [88]. The 166 kDa protein plays a role in the induction of apoptosis and is expressed as a long (DEFCAP-L) and active variant and as a 44 amino acid shorter (DEFCAP-S) and inactive variant (Figure 1a) [87]. DEFCAP-S lacks exon 14 of the NLRP1 gene, is ubiquitously expressed and defective in caspase-1 activation [48]. The function of DEFCAP-S/NLRP1Δex14 is not known. Roles of NLRP1 in apoptosis were later confirmed. An interaction of the sonic hedgehog (SHH) receptor Patched (Ptc) with DRAL/FHL2 (down-regulated in rhabdomyosarcomas LIM-domain protein/Four-and-a-half LIM protein 2) and NLRP1 or CARD8 [89] induces caspase-9-dependent apoptosis [90]. Furthermore, in the absence of caspase-1, NLRP1 can induce apoptosis requiring expression of ASC and caspase-8 [91]. NLRP1 is broadly expressed but at particularly high levels by neurons [92] and, interestingly, by differentiated epithelial cells, such as epidermal keratinocytes [88].

In addition to NLRP1, human primary keratinocytes (HPKs) express other inflammasome components such as ASC and caspase-1, as well as proIL-1β and -18 [93]. Most importantly, UVB radiation induces inflammasome activation in HPKs [93] and it is believed that this pathway underlies the induction of sunburn in humans [94]. UV radiation represents a major challenge and stressor for the skin and particularly for epidermal keratinocytes, which absorb the majority of UVB, whereas UVA is able to penetrate deeper into the skin. UVB does not only induce sunburn but is also an important inducer of skin cancer, the most frequent type of human malignancy [95,96]. UVB represents a classical inducer of apoptosis, particularly in keratinocytes, and this cell death pathway antagonizes the development of skin cancer [97,98]. Interestingly, caspase-1 is also a regulator of UVB-induced apoptosis of HPKs independently on inflammasome activation [39,99] and a role of caspase-1 in apoptosis is also discussed in other cell types [100,101,102,103]. So far, apart from *Toxoplasma gondii* in monocytic cells [104], UVB radiation represents the only known physiological activator of NLRP1 in human cells. By siRNA/shRNA approaches, NLRP1 but also NLRP3 were identified to sense UVB in human keratinocytes [93,105]. However, NLRP1 is considered the principal inflammasome sensor in human keratinocytes in vitro and in the skin [43]. Most likely, siRNA causes a kind of priming of keratinocytes for NLRP3 expression, as a study based on knockout HPKs generated by the CRISPR/Cas9 approach revealed that NLRP1 rather than NLRP3 is the UVB sensor in HPKs [106]. Moreover, the microbial toxin nigericin, a classical inducer of the NLRP3 inflammasome, is also sensed by NLRP1 in HPKs [106,107]. The molecular mechanisms underlying UVB-induced NLRP1 inflammasome activation are poorly understood but an increase in intracellular Ca^2+^ and reactive oxygen species (ROS) are likely contributing factors [93,99].

Upon treatment of HPKs with IFNγ or TNFα (priming), expression of other inflammasome sensors is induced [106,108]. Next to NLRP1, the AIM2 inflammasome also has relevant functions in human keratinocytes, since it senses infection by Herpes simplex virus (HSV) [109]. Moreover, the AIM2 inflammasome plays an important role in psoriasis, a common inflammatory skin disease, as cytosolic DNA triggers AIM2 inflammasome activation in HPKs and in lesion of patients suffering from the disease [110].

Although caspase-1 expression is required for the induction of UVB-induced skin inflammation in mice [93], UVB irradiation of murine keratinocytes neither in vitro nor in vivo induces inflammasome activation in keratinocytes [111]. Moreover, in contrast to human keratinocytes, murine keratinocytes do not express significant amounts of proIL-1β and NLRP1 [111]. Therefore, it is likely that an unidentified (immune) cell type in murine skin acts as the IL-1β producer upon UVB-irradiation, a duty that might be fulfilled by keratinocytes in human skin. These results demonstrate important differences between human and murine epidermis and keratinocytes. These differences in response to UVB and activation of NLRP1 suggest that mice represent a model with limited relevance for NLRP1-dependent UVB responses in human skin, such as sunburn and UVB-induced skin cancer development (see below). In contrast, there is evidence that AIM2, caspase-1 and proIL-1β are required for inflammatory memory of murine epithelial stem cells, sensitizing them for more efficient repair after tissue injury [112]. It should be considered that results obtained with keratinocytes in vitro may not be necessarily representative for keratinocytes in vivo. It might be that keratinocytes in culture and in vivo are in differently primed states concerning inflammasome activation. In conclusion, in contrast to murine keratinocytes, there is convincing evidence for expression and activation of the NLRP1 inflammasome in human keratinocytes.

## 6. The NLRP1 Inflammasome in Human Skin

Mutations of *NLRP1* have been described in patients who suffer from different inflammatory diseases with a skin phenotype, demonstrating that NLRP1 plays a particularly important role in the skin (Figure 3 and Table 1).

MSPC (multiple self-healing palmoplantar carcinoma) and FKLC (familial keratosis lichenoides chronica) are two rare monogenic inflammatory skin diseases, which are caused by gain-of-function (GoF) mutations of *NLRP1* [43]. The mutations occur in the aminoterminal pyrin domain of NLRP1 (MSPC) or in the LRRs (FKLC). NLRP1 is the only inflammasome sensor detectable in human skin, expressed by fibroblasts and keratinocytes [43]. Keratinocytes but not fibroblasts express major levels of caspase-1, ASC, proIL-1β and proIL-18, suggesting that keratinocytes but not fibroblasts express a functional NLRP1 inflammasome [43,111]. Most importantly, immortalized keratinocytes expressing the NLRP1 GoF mutants secreted more mature IL-1β than cells expressing wild type NLRP1 [43]. Furthermore, HPKs from MSPC and FKLC patients secreted more IL-1β and -18 than cells from healthy individuals [43]. Although a formal proof is missing, these results strongly suggest that MSPC and FKLC patients suffer from skin inflammation due to the expression of GoF mutants of NLRP1 in epidermal keratinocytes. Interestingly, patients suffering from CAPS, which is caused by GoF mutations of *NLRP3* (under normal conditions NLRP3 is not detectable in keratinocytes [43,106]), do also suffer from—among other symptoms—a skin phenotype [114]. MSPC predisposes patients to the development of SCCs (squamous cell carcinomas) [43], a major type of skin cancer, suggesting that NLRP1-dependent skin inflammation supports the development of tumors [115,116,117]. In line with this hypothesis is the observation that canakinumab, a monoclonal antibody targeting IL-1β, prevents the development of lung cancer [118,119]. In contrast, inflammasomes might also be implicated in the suppression of cancer development due to their roles in cancer cell death, repair processes, and anti-cancer immunity [115,116,117]. The finding that expression of the NLRP1 inflammasome, although GoF mutants of *NLRP1* predispose for SCC development [43], is suppressed in SCC cell lines and existing SCC tumors [113] supports a dual, paradoxical and cell type, stage- and/or time-specific role of inflammasomes in (skin) cancer development.

Recently, a novel autoimmune-inflammatory disease termed NAIAD (NLRP1-associated autoinflammation with arthritis and dyskeratosis) with a clear skin phenotype has been described. It is caused by mutations of *NLRP1* occurring either between the NACHT and LRRs or within the FIIND, close to the autoprocessing site [45]. The serum of patients contained higher levels of caspase-1, IL-18 and partially IL-1β, suggesting that the mutations cause increased NLRP1 activation.

Furthermore, several other SNPs of *NLRP1* are associated with an increased risk for the development of several diseases with partial involvement of the skin [40]. Vitiligo is a non-curable, autoimmune-linked disease causing depigmentation of the skin, hair, and oral mucosa due to the progressive loss of melanocytes [120]. Several different variants of the *NLRP1* gene were found to be associated with vitiligo [44,121,122,123]. The risk of contracting psoriasis is also associated with variants of the *NLRP1* gene [47,121]. Up to 2% of the population suffer from this condition, an inflammatory and hyper-proliferative skin disease, which can also affect other organs [124]. Other conditions with, but also without, an involvement of skin are linked to NLRP1. These include systemic lupus erythematosus [121,125], leprosy [126], juvenile-onset recurrent respiratory papillomatosis (JRRP) [46], and several others [40,127].

In conclusion, NLRP1 is the only inflammasome sensor expressed by keratinocytes in vitro and in vivo in the skin under normal and homeostatic conditions [43]. Furthermore, these cells express all the components required for NLRP1 inflammasome activation and the induction of inflammation [43,93]. As UVB [93] and talabostat [71] activate the NLRP1 inflammasome in human keratinocytes, it is likely that GoF mutations of *NLRP1* cause skin inflammation via hyperactivation of the inflammasome in epidermal keratinocytes. This demonstrates a particularly important role of NLRP1 in keratinocytes and in the skin. 

## 7. Outlook and Open Questions

Pharmacological targeting of NLRP1 activation represents an attractive strategy for treatment of the many patients suffering from NLRP1-dependent skin conditions. Due to its accessibility, the skin is predestined for topical application of small hydrophobic molecules able to penetrate into the tissue and into epidermal keratinocytes as well as skin-resident immune cells. Moreover, as patients with GoF mutations of *NLRP1* have mainly a skin phenotype, systemic treatment of patients suffering from NLRP1-dependent skin conditions with NLRP1 targeting molecules also might perhaps represent a possible approach with tolerable side-effects. Inhibition of NLRP1 might be effective for patients suffering from inflammatory skin conditions such as vitiligo or psoriasis. Furthermore, as the NLRP1 pathway is silenced in skin cancer [113], pharmacological activation of NLRP1 in affected skin lesions, for example by talabostat, might represent a successful approach for the treatment of skin cancer patients. The immune system plays a particularly important role in the development and treatment of skin cancer [96], and its activation, for example by imiquimod, is a successful strategy to combat skin cancer [128].

Several open points revolve around the molecular mechanisms underlying talabostat-induced inflammasome activation and particularly the cell type-specific consequences. What are the DPP8/9 substrates that regulate activation of human NLRP1, CARD8, and murine NLRP1b? Why does NLRP1 sense talabostat in keratinocytes, whereas CARD8 expression is required for caspase-1 activation in AML cells, although both cell types express NLRP1 and CARD8? Talabostat-induced CARD8 and downstream caspase-1 activation (and murine NLRP1b activation [51,69]) causes only pyroptosis [76], whereas talabostat-induced NLRP1 activation in human keratinocytes induces very effective processing of proIL-1β/-18 and secretion of the mature cytokines [71]. What determines the substrate specificity of caspase-1? Are ASC specks required for caspase-1-induced processing of proIL-1β and -18 but not for activation of gasdermin D [74,75]?

CARD8 is not only able to cause caspase-1 activation (in talabostat-treated AML cells) but can also inhibit NLRP3 upon direct interaction [82,83,84]. The active carboxyterminal effector domain of NLRP1 shares significant homology with the carboxyterminal fragment of CARD8. Can NLRP1 also inhibit NLRP3 activation? Is there a crosstalk between different inflammasome sensors? The NLRP1 variant lacking exon 14 (DEFCAP-S) cannot cause caspase-1 activation and is widely expressed [87]. What is its physiological role? Interestingly, a viral homologue of NLRP1 expressed by Kaposi’s sarcoma-associated herpesvirus (KSHV) binds and inhibits NLRP1, NLRP3, and NOD2 [129].

Since its discovery almost 20 years ago, we learned a lot about the molecular mechanisms underlying upstream and downstream pathways of NLRP1 and its physiologic roles in humans and mice. NLRP1 activation is regulated in a very complex manner, sharing certain aspects with other inflammasome sensors. In addition, NLRP1 activation is characterized by very specific features, such as proteolytic self-activation and functional degradation. In the near future, we can expect a better understanding of the mechanisms governing NLRP1 activation and its species-specific roles in the skin and beyond.

## Figures and Tables

**Figure 1 ijms-21-04788-f001:**
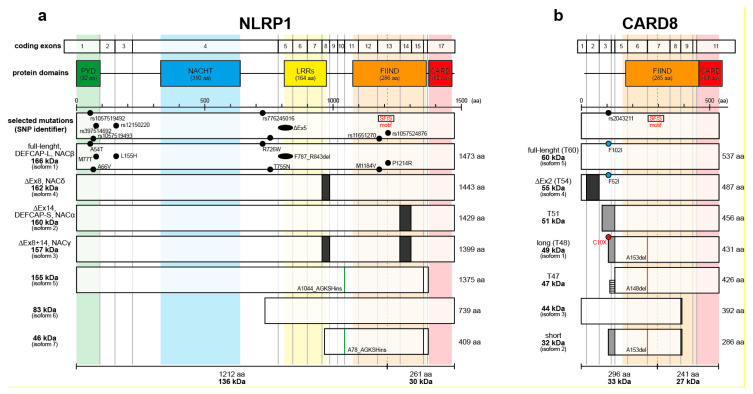
Isoforms of human NLRP1 and CARD8. (**a**) Full-length NLRP1 consists of five domains, an aminoterminal pyrin domain (PYD), a NACHT domain (NACHT), a leucine-rich repeats domain (LRRs), a function-to-find domain (FIIND), and a carboxyterminal caspase activation and recruitment domain (CARD). Self-processing occurs at S1213 in the SF/S motif of the FIIND. NLRP1 is encoded by up to 17 exons. Exon 8 is missing in isoform 4 (NACδ) and isoform 3 (NACγ) whereas exon 14 is missing in isoform 2 (DEFCAP-S, NACα) and isoform 3 (NACγ). Isoform 5 and 7 have 4 amino acids (GKSH) insertion after A1044 or A78, respectively. Selected NLRP1 mutations in human disease with skin involvement are depicted (for more information refer to Table 1). (**b**) CARD8 contains two domains, a FIIND and a CARD, with a variable aminoterminus and is encoded by up to 11 exons. T54 isoform misses exon 2 whereas isoform 1 (T48), isoform 2 (short), and T51 have alternative splicing replacing exons 1-3. CARD8 short, long and T47 have a deletion at A153 and A148, respectively. The SNP rs2043211 causes mutations in T60, T54, and T48, introducing in the latter a stop codon. Alternative splicing with translation of part of an intron generates a T47 isoform, which misses the nonsense mutation found in T48. Most of the variants of NLRP1 and CARD8 are poorly characterized.

**Figure 2 ijms-21-04788-f002:**
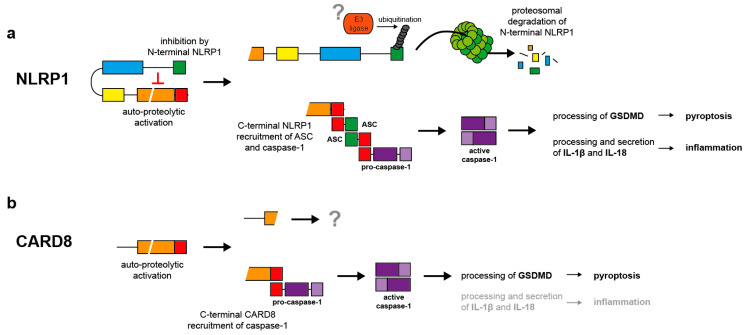
Mechanisms of NLRP1 and CARD8 activation. (**a**) NLRP1 activation is initiated by self-processing in the FIIND. However, the aminoterminal fragment inhibits the carboxyterminal effector protein containing the CARD by a non-covalent interaction. Upon ubiquitination of the aminoterminal inhibitory fragment by unknown ubiquitin ligases, the latter is degraded by the proteasome. The liberated effector fragment induces ASC speck formation, followed by caspase-1 activation and in turn processing of proIL-1β and -18 and gasdermin D (GSDMD). Secretion of the cytokines induces inflammation supported by GSDMD-induced pyroptosis. (**b**) Activation of CARD8 is poorly understood. Most likely, self-processing induces direct activation of caspase-1 without participation of ASC. In contrast to NLRP1, CARD8-induced caspase-1 activation causes mainly pyroptosis by GSDMD cleavage rather than processing of proIL-1β and -18.

**Figure 3 ijms-21-04788-f003:**
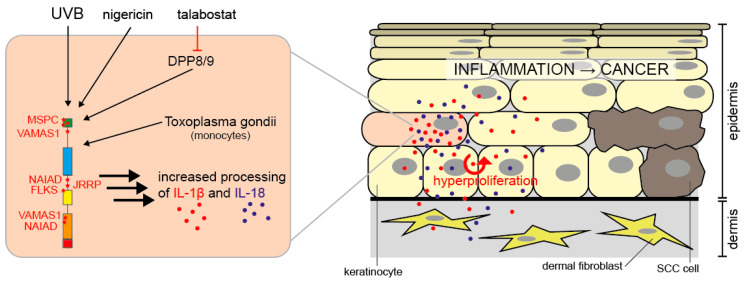
NLRP1 activation in human keratinocytes. In human keratinocytes, NLRP1 is activated by UVB radiation, nigericin, or talabostat. *Toxoplasma gondii* activates NLRP1 in monocytes. Via caspase-1 activation, this causes processing and secretion of proIL-1β and -18. IL-1β and -18 induce hyperproliferation of epidermal keratinocytes by autocrine and paracrine signaling and inflammation. Chronic NLRP1-dependent inflammation in the skin predisposes patients to the development of SCCs, a type of keratinocyte-derived skin cancer [43,71,106,113].

**Table 1 ijms-21-04788-t001:** Selected NLRP1 mutations in human disease with skin involvement.

Disease	SNP Identifier/ ClinVar	Mutation (Region/Domain)	Literature
**MSPC** (multiple self-healing palmoplantar carcinoma)	rs1057519492 / RCV000416558	A54T (Ex1, PYD)	[43]
	rs1057519493 / RCV000416502	A66V (Ex1, PYD)	
	rs397514692 / RCV000043505	M77T (Ex1, PYD)	
**VAMAS1** (vitiligo-associated multiple autoimmune disease susceptibility 1)	rs12150220 / RCV000004380	L155H (Ex3)	[44]
	rs11651270	M1184V (Ex13, FIIND)	
**NAIAD** (NLRP1- associated autoinflammation with arthritis)	rs776245016 / RCV000445355	R726W (Ex4)	[45]
	rs1057524876 / RCV000445359	P1214R (Ex13, FIIND)	
**FKLC** (familial keratosis lichenoides chronica)	RCV000445352	F787_R843del (ΔEx5, LRRs)	[43]
**JRRP** (juvenile-onset recurrent respiratory papillomatosis)	RCV001027400	T755N (Ex4)	[46]
**Psoriasis**	rs8079034C	- (intronic)	[47]
	rs878329C	- (5’-UTR)	

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
