# Peer review of "The NLRP1 Inflammasome in Human Skin and Beyond"

_ijms, 2020, doi:10.3390/ijms21134788_

Round 1

Reviewer 1 Report

Manuscript ID: ijms-852514

Type of manuscript: Review

Title: The NLRP1 inflammasome in human skin and beyond

Authors: Gabriele Fenini et al

Fenini and colleagues have provided an extensive and comprehensive review of NLRP1 which will be a valuable addition to the literature because of its synthetic approach connecting the basics of inflammasome action, relevance to epidermal hyperplastic/inflammatory disease, and possible eventual drug targeting of NLRP1.  As such, it will be well-received by those already in any of those fields.  As a minor point, it might prove to be a slow read for those looking to move into the research area.  For instance, section 6 makes clear a major take-away point (“In conclusion, ….).  It could be helpful to a more-general readership of a journal of broad scope if such capsule points were briefly added to some of the previous sections.  Given the expertise of the authors, if these were added, they need not require re-review of the manuscript. 

Specific points:

Page 1 – Page 2 as example

Is there an intended distinction between phrasing of “inflammasome sensor” and “inflammasome receptor”?

Page 3 – line 24

Please reference or clarify phrasing (such as in legend for Figure 1), seemingly equating SNP and mutation such as “common SNP …. causes mutations” as SNP as most often described with  “no effect on health or development” (https://ghr.nlm.nih.gov/primer/genomicresearch/snp_.  Cited references (such as Cell. 2016 Sep 22;167(1):187-202.e17) appear to make a distinction with phrasing including “disease-causing mutants or common NLRP1 SNPs”. 

Page 3

Minor point: Although the manuscript layout in journal form helped to integrate text reading with the figures, there is some concern on font sizes in labeling:  Figure 1 (especially with the left of the figure such as for “selected SNPs and natural variants”) and Figure 2 and 3 (protein subdomains).

Page 8 – Figure 3

Please label cell types (such as apparent dermal fibroblasts) and clarify implied biology as to which cells of the stratified epithelium are undergoing “hyperproliferation”.  The diagram seems to indicate these would be supra-basal cells.  Call out the apparent color-coding of IL-1b and IL-18 from the expanded cell and the secretion occurring in the stratified diagram.  Are the brown/grey cells at the right of the diagram supposed to be the genesis of SCC? Please label or indicate in legend. 

Page 8 – line 18

Minor point: “prove” or proof?

Page 9 – lines 18-20

Although topical applications seem reasonable, systemic treatment may be overly optimistic given the wide-spread tissue distribution of NLRP1.  This might be phrased more cautiously. 

Author Response

Fenini and colleagues have provided an extensive and comprehensive review of NLRP1 which will be a valuable addition to the literature because of its synthetic approach connecting the basics of inflammasome action, relevance to epidermal hyperplastic/inflammatory disease, and possible eventual drug targeting of NLRP1.  As such, it will be well-received by those already in any of those fields.  As a minor point, it might prove to be a slow read for those looking to move into the research area.  For instance, section 6 makes clear a major take-away point (“In conclusion, ….).  It could be helpful to a more-general readership of a journal of broad scope if such capsule points were briefly added to some of the previous sections.  Given the expertise of the authors, if these were added, they need not require re-review of the manuscript.

Our reply: We thank the reviewer for carefully reading and appreciation of our manuscript. Furthermore, we thank the reviewer for the advice to add one or more a take-away sentences at the end of sections. We followed this advice as can be seen in the new version of our manuscript.

Specific points:

Page 1 – Page 2 as example

Is there an intended distinction between phrasing of “inflammasome sensor” and “inflammasome receptor”?

Our reply: No, we used “sensor” and “receptor” as synonym. In the new version, we replace “receptor” by “sensor” and we included the definition of NOD-like receptor (NLR).

Page 3 – line 24

Please reference or clarify phrasing (such as in legend for Figure 1), seemingly equating SNP and mutation such as “common SNP …. causes mutations” as SNP as most often described with  “no effect on health or development” (https://ghr.nlm.nih.gov/primer/genomicresearch/snp_.  Cited references (such as Cell. 2016 Sep 22;167(1):187-202.e17) appear to make a distinction with phrasing including “disease-causing mutants or common NLRP1 SNPs”.

Our reply: We thank the reviewer for this important hint and changed the text legend in Figure 1.

Page 3

Minor point: Although the manuscript layout in journal form helped to integrate text reading with the figures, there is some concern on font sizes in labeling:  Figure 1 (especially with the left of the figure such as for “selected SNPs and natural variants”) and Figure 2 and 3 (protein subdomains).

Our reply: We changed the font sizes in the Figures 1 and 3.

Page 8 – Figure 3

Please label cell types (such as apparent dermal fibroblasts) and clarify implied biology as to which cells of the stratified epithelium are undergoing “hyperproliferation”.  The diagram seems to indicate these would be supra-basal cells.  Call out the apparent color-coding of IL-1b and IL-18 from the expanded cell and the secretion occurring in the stratified diagram.  Are the brown/grey cells at the right of the diagram supposed to be the genesis of SCC? Please label or indicate in legend.

Our reply: Thanks you, we did so and this clearly improved the Figure.

Page 8 – line 18

Minor point: “prove” or proof?

Our reply: Has been corrected.

Page 9 – lines 18-20

Although topical applications seem reasonable, systemic treatment may be overly optimistic given the wide-spread tissue distribution of NLRP1.  This might be phrased more cautiously.

Our reply: The reviewer is right and we changed the sentence.

Reviewer 2 Report

This is an excellent review of the current understanding of NLRP1 regulation in human keratinocytes.  The authors list on p 8 that other cells highly express NLRP1 and include fibroblasts in this list.  However, since the title reads “and beyond”, it would be of interest to list the expression of NLRP1 in neurons.  There is ample evidence that NLRP1 controls caspase 1 and caspase 6 in neurons.  The author may want to include a discussion of the differences in regulation of NLRP1 in neurons and keratinocytes.

Minor point:

  1. 8 line 18. Prove should read: proof

Author Response

This is an excellent review of the current understanding of NLRP1 regulation in human keratinocytes.  The authors list on p 8 that other cells highly express NLRP1 and include fibroblasts in this list.  However, since the title reads “and beyond”, it would be of interest to list the expression of NLRP1 in neurons.  There is ample evidence that NLRP1 controls caspase 1 and caspase 6 in neurons.  The author may want to include a discussion of the differences in regulation of NLRP1 in neurons and keratinocytes.

Our reply: We thank the reviewer for carefully reading and appreciation of our manuscript. In the new version of the manuscript, we mention that NLRP1 is also highly expressed by neurons (page 7). However, we do not want to discuss the role of NLRP1 in neurons or other cell types than keratinocytes. “Beyond” in the title refers to CARD8.

Minor point:

18.8 line 18. Prove should read: proof

Our reply: We corrected the mistake.